# Study on the Measurement of the On-Site Overvoltage and Internal Temperature Rise Simulation of the EMU Arrester

Shenghui Wang [1],*, Qi Ou [1], Shengfeng Lei [1], Huaqi Liu [1], Shuaitao Mao [1], Qizhe Zhang [2], Jian Liu [3] and Fangcheng Lv [1]

1 State Key Laboratory of Alternate Electrical Power System with Renewable Energy Sources, North China Electric Power University, Beijing 102206, China
2 China Elect Power Res Inst, Beijing 100192, China
3 CRRC Changchun Railway Vehicles Co., Ltd., Changchun 130011, China
* Correspondence: hdwsh@ncepu.edu.cn; Tel.: +86-189-1229-9702

**Abstract:** In order to analyze the explosion accidents of the CRH5 EMU roof arrester in recent years, an internal temperature measuring platform based on fluorescence fiber was established, and the temperature distribution characteristics under the continuous operating voltage and high-current impulse were analyzed. The test results show that passing section overvoltage and steep impulse overvoltage have higher amplitudes, while high-harmonic overvoltage has a lower amplitude but a longer duration. The maximum temperature rise of the arrester was 5.2 °C under 34 kV for 3 h. The surface temperature of the valve plate column was high in the middle and low on both sides; the maximum temperature difference at different positions was only 2.2 °C. The maximum temperature of the valve plate column rose to 97.6 °C under 105 times of the high-current impulse, and the maximum temperature difference at different positions reached 33.8 °C. Then, the actual overvoltage of the arrester in operation was measured and analyzed statistically, and the arrester simulation model was established. The temperature characteristics of the normal arrester and the arrester with the electric tree were studied under the actual typical overvoltage, and the influence of air velocity on the internal temperature rise was analyzed. The simulation results show that, due to the low amplitude and small current of high-harmonic overvoltage, the internal temperature rise of the normal and defective arresters was very small. Under the effects of passing section overvoltage and steep impulse overvoltage, the internal temperature of the normal arrester can reach 36.57 °C and 241 °C, and the arrester with the electric tree defect can reach 44.75 °C and 536 °C, respectively. The air velocity has little effect on the internal temperature rise of the arrester. Passing section overvoltage and steep impulse overvoltage occur frequently and have an obvious influence on the internal temperature rise of the arrester, so the roof overvoltage of the EMU is an important reason for the arrester burst.

**Keywords:** EMU; arrester; temperature distribution; simulation; electric tree

## 1. Introduction

The roof arrester is an important piece of overvoltage protection equipment of the Electrical Multiple Unit (EMU) [1–3]. Due to the complex operating environment, its performance will gradually deteriorate or even form burst accidents [4–6]. According to statistics, from 2017 to 2021, a total of 12 arrester burst faults occurred in China's CRH5 EMUs. For example, the pressure-release vent of the arrester was burst on 21 December 2019. The on-site accident picture is shown in Figure 1.

The internal temperature of the arrester may rise sharply under some types of overvoltage, leading to thermal collapse. In order to analyze the influence of overvoltage on the internal temperature rise of the arrester and whether it will cause the thermal collapse of the arrester, it is necessary to measure the on-site overvoltage waveform and study the internal temperature rise of the arrester so as to help explain the cause of the roof arrester burst.

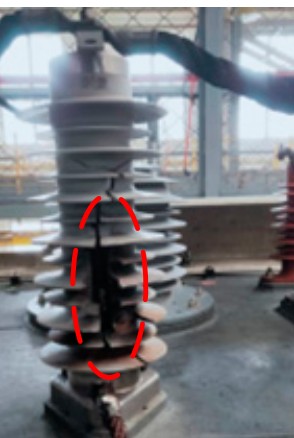

**Figure 1.** The scene of the accident.

In view of the causes and characteristics of roof overvoltage and the thermal characteristics of the arrester, relevant scholars have carried out some studies. According to the theoretical analysis of the literature [7,8], the pantograph is prone to the arc phenomenon when the EMU passes through phase separation, which will cause a sudden voltage increase and form a trapezoidal pulse. In the literature [9], the waveform of high-harmonic overvoltage was obtained from the field-measured data. The oscillation duration of overvoltage can be up to 2 min, and when the overvoltage amplitude is high, the roof arrester will burst. Since the metal oxide arrester (MOA) does not have clearance, the leakage current will flow through the valve plate under normal voltage, which will result in power loss and the valve plate heating [10–15]. The literature [16] pointed out that the leakage current under overvoltage was the direct cause of the heating. In the literature [17], through the disassembly and inspection of the arrester, it was found that the causes of its abnormal heating are internal dampness, the aging of the valve plate and the contamination of the external insulation. In the literature [18], the internal hot spot and maximum temperature can be estimated by measuring the external temperature of the arrester. The literature [19] pointed out that the energy absorption process of MOA under the action of impulse current is an adiabatic temperature rise process. When the temperature is high enough, the material properties will change or even burn out. The literature [20] analyzed the temperature of the arrester through simulation and found that when the valve plate is damaged, its operating state can be judged by temperature. The literature [21] used COMSOL to establish the temperature field calculation model of the arrester and found that the internal temperature rise of the defective arrester was significantly higher than the external surface temperature rise, and there was a partial overheating phenomenon.

Based on the above research status, it can be seen that the current research mainly focuses on the analysis of a single type of overvoltage waveform and the internal temperature rise of the arrester under the action of the working voltage. At present, there is still a lack of on-site overvoltage measurements and comprehensive statistical analyses of the roof overvoltage waveforms, along with a lack of research on the internal temperature rise of the arrester under the action of on-site measured overvoltage. Therefore, in this paper, experiments and simulation based on on-site measured overvoltage were carried out to study the relationship between overvoltage and the internal temperature rise of the arrester and conduct a comprehensive statistical analysis of the on-site-measured overvoltage, which can explain the relationship between the roof overvoltage and the arrester burst. Firstly, the internal temperature test of the arrester under the action of power frequency overvoltage and high-current impulse was designed, and the internal temperature rise characteristics of the arrester under the action of overvoltage were obtained. In order to acquire the actual overvoltage waveform required by the simulation, an online monitoring system of roof overvoltage was established to acquire the overvoltage waveforms suffered by the EMU in the actual operation and analyze the statistical characteristics of the typical overvoltage

waveforms. Finally, the electrothermal coupling model of the arrester was established, and the internal temperature rise characteristics of the arrester under different conditions were obtained under the action of actual overvoltage. The research results have a certain guiding significance for explaining the cause of roof arrester bursts and the selection of arresters.

## 2. Arrester Temperature Rise Test

### 2.1. The Tested Sample

For the convenience of measuring the temperature rise inside the arrester, a customized test arrester was used to carry out the subsequent test in this paper. The composite insulation material includes silicone rubber and epoxy glass fiber pipe; the basic structure is consistent with that employed in engineering practice. Five temperature detection holes are arranged on the arrester, which go through the entire insulated cylinder, and the high-precision fluorescent optical fiber was placed inside the customized arrester, as shown in Figure 2.

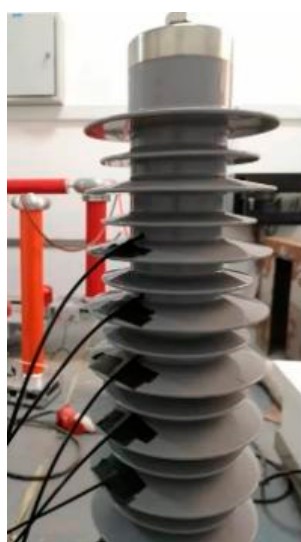

**Figure 2.** Customized arrester structure.

In Figure 2, the hole diameter is 6 mm and the spacing is 56 mm. The total height of the arrester is 520 mm, the rated voltage is 27.5 kV, the continuous operating voltage is 34 kV, the nominal discharge current is 10 kA, the $U_{1mA}$ is 61.4 kV, the leakage current under 0.75 times of $U_{1mA}$ is 14 μA, the residual voltage of the lightning impulse current is 104.3 kV, and the power frequency reference voltage is 42.6 kV.

### 2.2. Fluorescent Optical Fiber Temperature Measuring System

The fluorescent optical fiber temperature measuring device is shown in Figure 3. It is composed of a fluorescent optical fiber temperature sensor, a temperature analyzer, a temperature display device and a computer.

The type of temperature analyzer is FA-09, and its resolution can reach 0.1 °C. The type of fluorescent fiber optic temperature sensor is FS-03, and its accuracy is ±0.3 °C. The fluorescence fiber is connected to channels 1, 2, 3, 4 and 5 of the temperature analyzer successively from the high-voltage side to the low-voltage side, and the temperature variation of each channel is recorded by the computer.

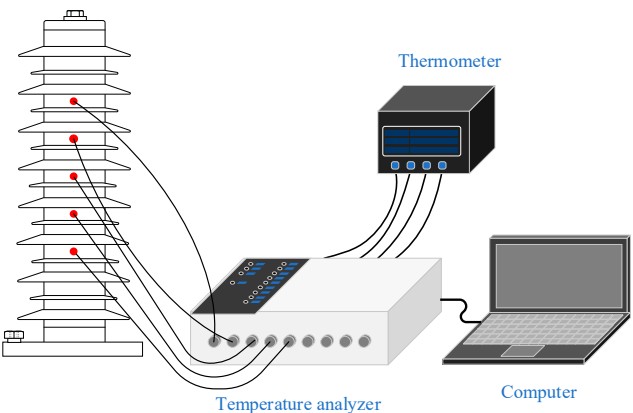

**Figure 3.** Fluorescent optical fiber temperature measuring system.

### 2.3. Test Method and Procedure

(1) The temperature rise test under continuous operation voltage. Before the test, the indoor ambient temperature was 25.8 °C. First, the 34 kV voltage was applied to the arrester for 3 h; then, the power was cut off and it was let to cool down to room temperature naturally; finally, the temperature changes in each channel were measured.

(2) The temperature rise test under high-current impulse. The impulse current was generated with a 1200 kV impulse voltage generator, which is shown in Figure 4. First, each stage capacitor of the impulse voltage generator was charged in parallel; then, the trigger device sent out an ignition, and the steep impulse current through the arrester can reach 10 kA. The next impulse current test was carried out after an interval of 60 s, with 15 shocks in each group and an interval of 300 s between the two adjacent groups.

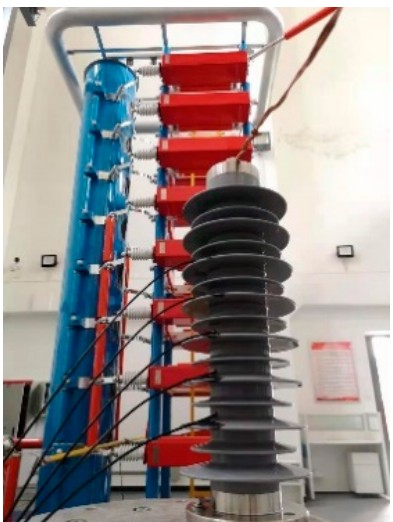

**Figure 4.** The impulse voltage generator.

### 2.4. Test Results and Analysis

2.4.1. Temperature Rise Analysis under Continuous Operating Voltage

Based on the above experimental study, the temperature distribution of different channels is shown in Figure 5.

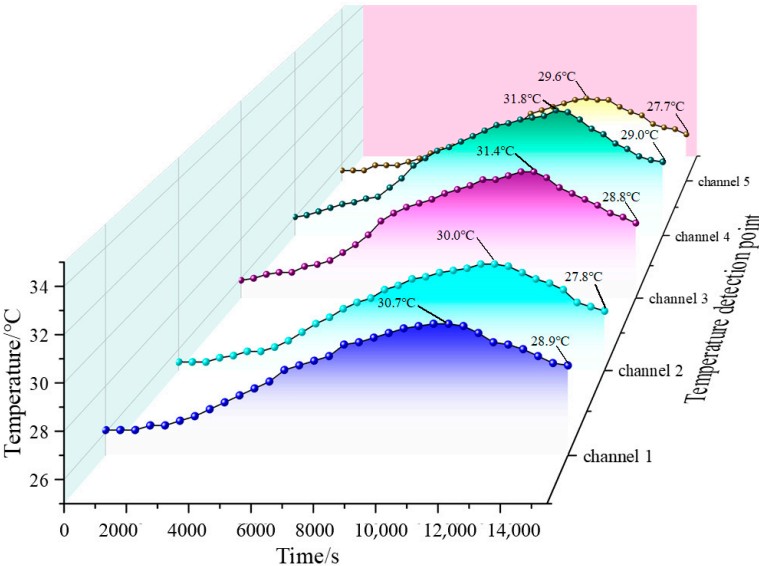

**Figure 5.** Temperature test of the arrester under continuous operation voltage.

The analysis of Figure 5 shows that the temperature distribution in each channel of the normal arrester is similar under the continuous operating voltage. During the first 3600 s, the temperature in each channel rises slowly, reaching its maximum value around 10,800 s. Channel 4 has the highest temperature; it can reach 31.8 °C. After the applied voltage was cut off, the temperature decreased slowly in each channel. Within 4700 s, it decreased by about 3 °C in channels 3 and 4 and decreased by about 2 °C in the rest.

The maximum temperature difference at the same position with different times was 5.2 °C, and that at different positions at the same time was 2.2 °C. Before and after the test, the temperature difference of each channel was not obvious; therefore, the threat of power frequency overvoltage to the safe operation of the EMU roof arrester is limited.

2.4.2. Temperature Rise Analysis under High-Current Impulse

In the impulse current experiment, the impulse current waveform generated by the impulse voltage generator and the residual voltage of the arrester is shown in Figure 6.

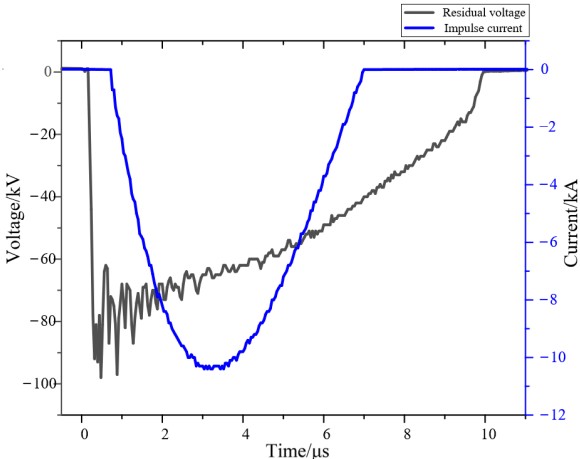

**Figure 6.** Impulse current and residual voltage waveform.

It can be seen from Figure 6 that the residual voltage amplitude is about 92 kV with an impulse current of 10 kA. As the temperature of the valve plate reaches 115 °C, it can accelerate the aging of the arrester. We stopped applying voltage and let it cool naturally

at this time, and then we observed the temperature changes in each channel, as shown in Figure 7.

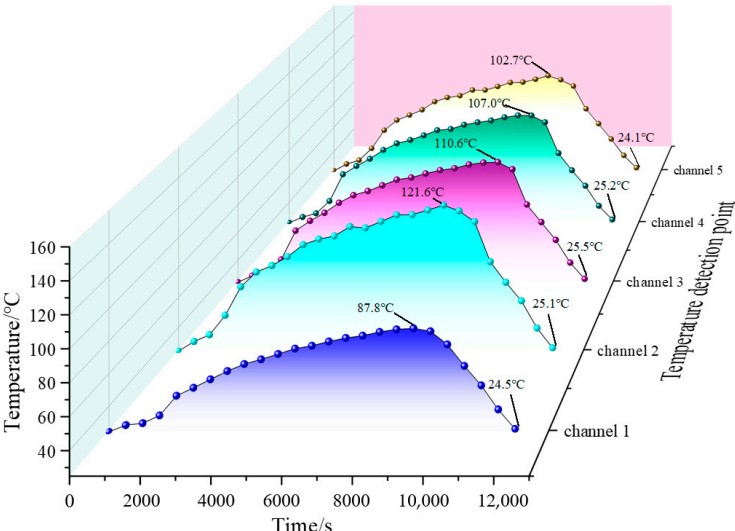

**Figure 7.** Arrester temperature test under impulse current.

Figure 7 shows that the temperature distribution in each channel under the impulse current is similar to the continuous operating voltage. Among the five channels, channel 3 reached the highest temperature at 8500 s, which was 121.6 °C, and that in channel 1 was the lowest, which was 87.8 °C. The temperature distribution was high in the middle and gradually decreased to both sides, which is due to the uneven voltage distribution and the heat dissipation of the metal at both sides of the valve column.

At the beginning stage of heating up, the temperature in each channel increased slowly. Subsequently, the temperature rise rates of channels 2, 3 and 4 were significantly accelerated, which may due to channels 1 and 5 being closer to the metal part and the heat conduction being faster than that in other channels. After the impulse experiment was stopped, the temperature overshoot was small and immediately began to decline slowly. From the declining trend, although channel 1 is close to the metal part, the temperature decrease rate is lower than that of the other channels; this is because a large amount of hot air gathered in the upper part, and there is only limited heat conduction through the metal end.

In the experiment, the maximum temperature difference of the arrester at the same position at different times is 97.6 °C, and that at different positions at the same time is 33.8 °C. Before and after the experiment, the valve plate temperature difference is large, and the highest temperature exceeded 115 °C. The arrester will accelerate the aging of the valve plate under this condition. The EMU often suffers from operating overvoltage and lightning overvoltage, and the arrester will absorb a large amount of energy when the overvoltage amplitude is too large or there are multiple overvoltage shocks in a short time. With the increase in active power loss, the internal temperature rises sharply and eventually leads to thermal collapse.

Based on the above study, under the continuous operation voltage, the energy absorbed by the arrester was small, the temperature increased by 5.2 °C within three hours and the temperature distribution of the internal valve plate was uniform. Under the action of high-current impulse, the energy absorbed by the arrester was large, the temperature rise reached 97.6 °C and the temperature varied greatly at different positions inside. Therefore, the study of the temperature distribution characteristics under the action of large impulse overvoltage is conducive to analyzing the cause of its faults. Since it is difficult to measure the internal temperature rise of the arrester under the action of actual overvoltage, the simulation method is adopted to carry out the relevant research.

## 3. Field Overvoltage Waveform Acquisition and Statistics

### 3.1. Field Overvoltage Waveform Acquisition

In order to make the calculation results of the simulation model more accurate, the overvoltage waveforms used in the model were the typical overvoltage waveforms suffered by the EMU in the actual operation. In order to acquire the typical overvoltage waveforms, an online monitoring system of roof overvoltage was established. The system consists of a pantograph, arrester, capacitive voltage divider, data acquisition card and computer. The model of the arrester is YH10WT-42/105. The low-voltage side signal of the capacitive voltage divider was introduced into the HS4 data acquisition card inside the EMU through coaxial cable, and its sampling rate was set to 100 kHz. The HS4 data acquisition card has four sampling channels, which can realize high-speed continuous synchronous acquisition. In this paper, the flow disk mode is used to achieve continuous signal acquisition and avoid data leakage, and real-time signal acquisition is stored in the computer.

The cumulative test distance was about 100,000 km, and 1274 groups of overvoltage waveforms were extracted from the original data. The overvoltage was classified into passing section overvoltage, high-harmonic overvoltage and steep impulse overvoltage according to their characteristics. The three typical overvoltage waveforms are shown in Figure 8.

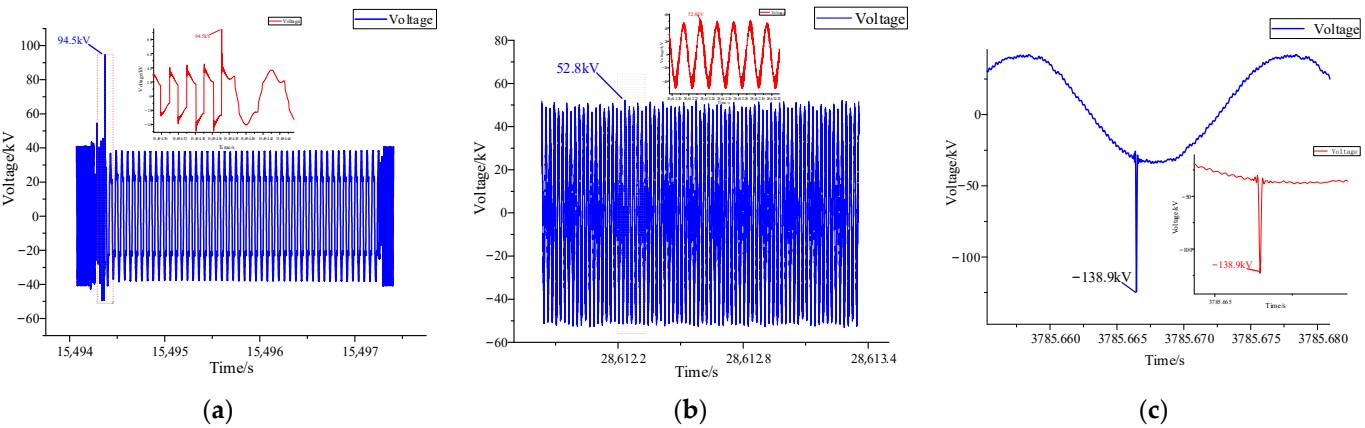

**Figure 8.** Three typical overvoltage waveforms. (**a**) Passing section overvoltage. (**b**) High-harmonic overvoltage. (**c**) Steep impulse overvoltage.

The passing section overvoltage is mainly caused by the action of the circuit breaker when the EMU passes the neutral section; its waveform is shown in Figure 8a. From Figure 8a, it can be seen that the voltage waveform of the arrester changed from a sinusoidal signal to a non-sinusoidal periodic signal and superimposed periodic impulse voltage. The maximum amplitude of impulse voltage is 94.5 kV, and the duration of a single impulse is about 0.2 ms.

Due to the existence of power electronic equipment such as the rectifier and inverter in the traction power supply system of the EMU, the voltage at both ends of the arrester contains high-harmonic components. When the harmonic content is high, the voltage peak may increase, as shown in Figure 8b. The voltage waveform was highly distorted, reaching a peak of 52.8 kV compared to 38.9 kV under normal conditions.

Steep impulse overvoltage has a higher amplitude and may occur continuously, which is likely to cause the arrester to heat up. The maximum amplitude of the steep impulse overvoltage detected during the EMU operation is −138.9 kV, as shown in Figure 8c. The duration of impulse voltage is about 0.14 ms. During the overvoltage occurrence period, there is no abnormal harmonic voltage or process of passing the neutral section.

In Figure 8, the passing section overvoltage and steep impulse overvoltage have higher amplitudes, reaching 94.5 kV and 138.9 kV, respectively, while the lightning shock residual voltage of the arrester under the nominal discharge current (10 kA) should be

less than 105 kV. If the arrester is attacked by overvoltage with such a high amplitude in a short period of time, the temperature of the valve plate will rise sharply, which may lead to thermal aging or the damage of the arrester. Therefore, it is necessary to study the temperature variation characteristics of the arrester under the action of overvoltage.

### 3.2. Statistical Analysis of the Overvoltage Waveform

Due to the randomness of overvoltage, it is necessary to analyze the amplitude distribution statistically. Through the statistics of the overvoltage waveforms, the overvoltage amplitude was classified into five ranges, as shown in Table 1. The occurrence times of different types of overvoltage are shown in Figure 9.

**Table 1.** Overvoltage amplitude distribution.

| Type | Overvoltage Amplitude Range (kV) | | | | |
| --- | --- | --- | --- | --- | --- |
| | 50–60 | 60–70 | 70–80 | 80–90 | >90 |
| Passing section overvoltage | 819 | 235 | 89 | 15 | 2 |
| High-harmonic overvoltage | 71 | 0 | 0 | 0 | 0 |
| Steep impulse overvoltage | 6 | 13 | 11 | 8 | 5 |

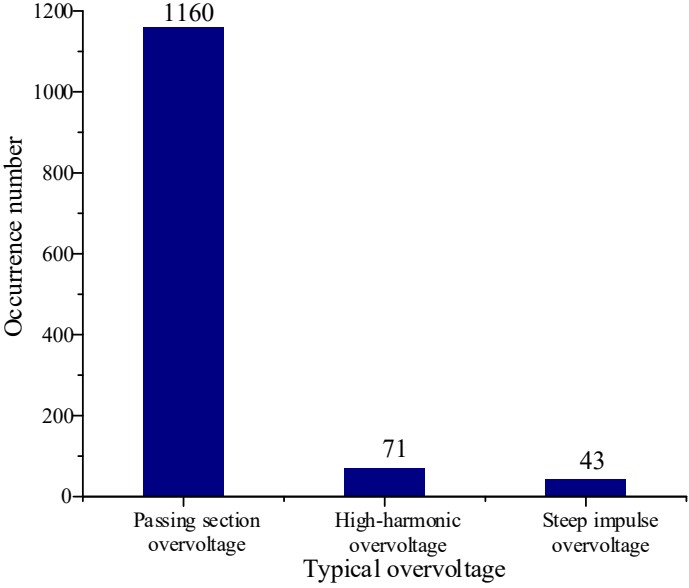

**Figure 9.** Number of three kinds of overvoltage.

The arrester's continuous operating voltage is 34 kV, and the corresponding voltage amplitude is 48 kV, that is, the voltage amplitude does not exceed 48 kV and poses almost no threat to the arrester. Therefore, voltage waveforms with amplitudes greater than 50 kV were counted. As can be seen from Table 1, most amplitudes of the passing section overvoltage are less than 70 kV, and very few overvoltage amplitudes exceed 90 kV. The amplitudes of high-harmonic overvoltage are all less than 60 kV. The amplitudes of steep impulse overvoltage are relatively high; this includes five overvoltage waveforms. Their amplitudes exceed 90 kV, and the maximum voltage amplitude is 138.9 kV. Therefore, in general, steep impulse overvoltage is a relatively great threat to the arrester.

Figure 9 shows that passing section overvoltage occurs most frequently, and steep impulse overvoltage occurs the least, but its amplitude is high, and multiple steep impulses may occur in one cycle.

## 4. Simulation Research on the Temperature Distribution of the Arrester

### 4.1. Simulation Model

In the above research, the internal temperature rise characteristics of the normal arrester under the action of power frequency overvoltage and high-current impulse had been obtained, and the typical overvoltage waveforms had been obtained. In order to further analyze the internal temperature rise characteristics of the arrester under the action of typical overvoltage under different circumstances, an electrothermal coupling simulation model on the basis of the actual size of the CRH5 EMU roof arrester was established with finite element simulation software, as shown in Figure 10.

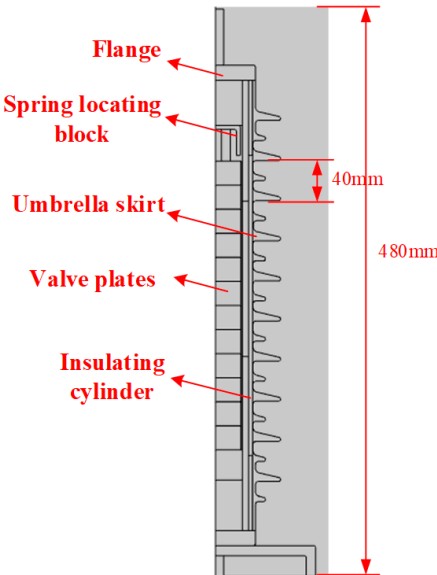

**Figure 10.** Arrester model.

The arrester model consists of a flange, sheath, spring locating block, valve plate, insulation cylinder, umbrella shed and pedestal. The height of each valve plate is 24 mm, and the radius is 27 mm. The material settings of each part of the model are shown in Table 2.

**Table 2.** Material parameter settings.

| Model Composition | Material | Relative Dielectric Constant | Conductivity/(S·m$^{-1}$) |
|---|---|---|---|
| Umbrella skirt, sheath | Silicone rubber | 2.3 | $1.0 \times 10^{-8}$ |
| The internal air gap of the arrester | Air | 1.0 | $1.0 \times 10^{-12}$ |
| Flange | Iron | $1.0 \times 10^{6}$ | $1.1 \times 10^{7}$ |
| Spring locating block | Aluminum | $1.0 \times 10^{6}$ | $3.8 \times 10^{7}$ |
| Valve plate | Zinc oxide | 420 | As shown in Table 3 |
| Insulation cylinder | Epoxy resin | 4.0 | $1.0 \times 10^{-8}$ |

**Table 3.** Conductivity calculation.

| Voltage/V | Current/A | Conductivity/(S·m$^{-1}$) |
|---|---|---|
| $2.58 \times 10^{4}$ | $3 \times 10^{-7}$ | $1.6 \times 10^{-9}$ |
| $4.08 \times 10^{4}$ | $4.6 \times 10^{-6}$ | $1.55 \times 10^{-8}$ |
| $5.52 \times 10^{4}$ | $9.01 \times 10^{-5}$ | $1.25 \times 10^{-7}$ |
| $6.25 \times 10^{4}$ | $7.4 \times 10^{-4}$ | $1.63 \times 10^{-6}$ |
| $6.3 \times 10^{4}$ | $1.01 \times 10^{-3}$ | $2.21 \times 10^{-6}$ |
| $8.39 \times 10^{4}$ | $2.0 \times 10^{3}$ | 3.28 |
| $1.01 \times 10^{5}$ | $1.01 \times 10^{4}$ | 32.46 |

### 4.2. Calculation of the Nonlinear Conductivity of Zinc Oxide

The conductivity of the metal oxide arrester (MOA) valve plate varies with the impulse voltage value, and the setting of the parameter value directly affects the accuracy of the simulation results [22–24]. In order to obtain the nonlinear resistance characteristic, the relation curve between conductivity and voltage has been studied by a relevant experiment.

First, the DC high voltage was applied to the arrester, and the voltage-ampere characteristics of the arrester with milliampere current were obtained. Then, a high-current impulse was applied, and the voltage-ampere characteristics with a large current were obtained. The measured curve is shown in Figure 11.

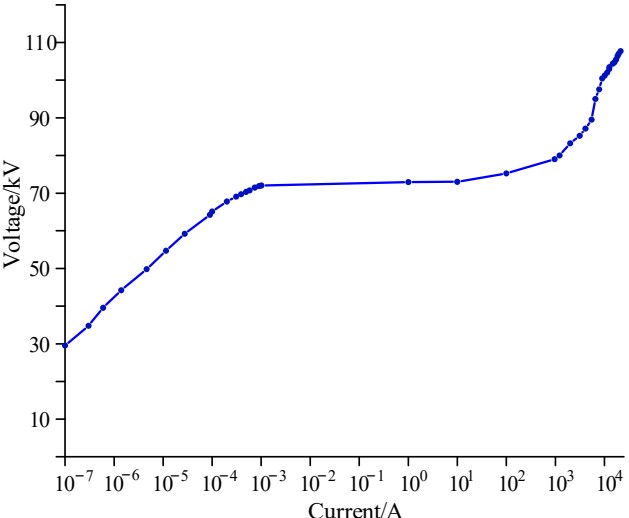

**Figure 11.** Volt-ampere characteristics of the arrester.

The conductivity of the arrester under different voltages can be calculated according to Formula (1).

$$\rho = \frac{R \cdot S}{H}, \sigma = \frac{1}{\rho} \tag{1}$$

where $\rho$ is the resistivity, $R$ is the resistance value, $S$ and $H$ are the cross-section area of the arrester valve plate part, respectively, and $\sigma$ is the conductivity of the valve plate. The partial calculation results are shown in Table 3.

### 4.3. Electrothermal Coupling Calculation Method

In the electrothermal coupling analysis, the finite element method has an excellent ability to simulate heterogeneous material properties and complex geometric boundaries. Therefore, it is very appropriate to use the finite element method to simulate and solve the temperature field of the arrester. The calculation steps are summarized as follows: (1) determine the governing equations and boundary conditions of the electric field and temperature field; (2) discretize the research field to obtain non-overlapping subdivision units; (3) calculate the difference of the unit field; (4) use the iterative method to solve the equations.

The electrothermal coupling adopts indirect coupling. First, the energy loss value of the valve plate column of the arrester was obtained by solving the electromagnetic field, and then the loss value was used as the heat source of the heat transfer to solve the temperature field.

#### 4.3.1. Governing Equations and Boundary Conditions

Due to the short duration of the overvoltage, the calculation of the temperature field in this paper is mainly based on the transient heat transfer differential equation, coupled with the electric field governing equation.

The governing equation of the electric field in the whole solution domain is as follows:

$$\nabla \cdot J = Q_{j,v}$$
$$J = \sigma E + \frac{\partial D}{\partial t} + J_e \tag{2}$$
$$E = -\nabla V$$

where $\sigma$ is the electrical conductivity, $D$ is the electric displacement, $J$ is the current density, $J_e$ is the external injection current density, $Q_{j,v}$ is the voltage source, $E$ is the electric intensity and $V$ is the electric potential.

The external boundary condition of the arrester is:

$$n \cdot J = 0 \tag{3}$$

where $n$ is the interface normal vector and $J$ denotes the current density.

The internal boundary condition of the arrester is:

$$n_2 \cdot (J_1 - J_2) = 0 \tag{4}$$

where $n_2$ is the interface normal vector and $J_1$ and $J_2$ are the current densities on both sides of the interface.

The heat transfer governing equation and boundary conditions are as follows:

According to the heat transfer theory, the governing equation of the transient heat transfer of the arrester is:

$$\rho C_p \frac{\partial T}{\partial t} + \rho C_p v \cdot \nabla T + \nabla \cdot q = Q + q_0 + Q_{\text{ted}}$$
$$q = -k \nabla T \tag{5}$$

where $\rho$ is the density of the material, $C_p$ is the heat capacity of the material, $T$ is the temperature gradient, $v$ is the velocity vector, $q$ is the rate of heat conduction, $k$ is the thermal conductivity of the material, $Q_{\text{ted}}$ is the increase in heat of the heat source, $q_0$ is the convective heat flux, $Q$ is the heating power per unit volume and its value is the dot product of $E$ and $J$ in the governing equation of the electric field.

The boundary conditions of the temperature field are as follows:

$$-n \cdot q = q_0$$
$$q_0 = h(T_{ext} - T) \tag{6}$$

where $n$ is the interface normal vector, $h$ is the heat transfer coefficient of the external surface of the arrester, $T_{ext}$ is the external boundary temperature and $T$ is the internal boundary temperature. According to relevant literature [25], the comprehensive heat transfer coefficient of the silicon rubber outer surface of the arrester is generally set as 4~10 W/(m$^2$·K). When there is no or little air flow, the convective heat transfer coefficient is between 4~10 W/(m$^2$·K). Therefore, the heat transfer coefficient is taken as 10 W/(m$^2$·K) in the simulation.

### 4.3.2. Meshing of the Simulation Model

In the simulation, the quadrilateral element was used to discretize the solution domain. Since the valve plate column was a heat source and the temperature changes greatly, the meshing of the internal valve column was divided by fine meshes, and the air domain around the valve column, insulating cylinder, etc. was also divided into fine meshes to avoid sharp angles and severe mesh gradients. The rest of the model was divided into coarse meshes to simplify the calculation. The meshed finite element model is shown in Figure 12.

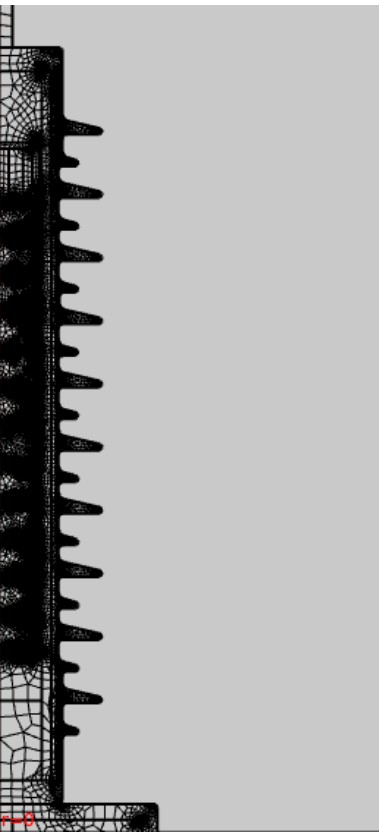

**Figure 12.** Meshing of the Simulation Model.

*4.4. Electric Tree Defect Setting*

According to the dismantling inspection results, the electric tree was found on the sides of the valve column in the burst arresters, as shown in Figure 13.

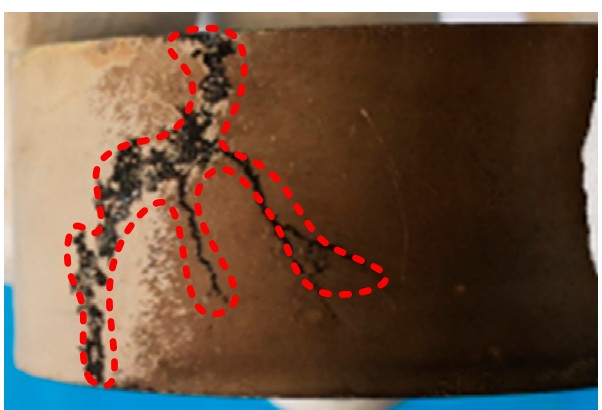

**Figure 13.** Defect valve plates.

It can be seen from Figure 13 that there is a penetration trace on the side of the valve plate. Once the electric tree is formed, it will short-circuit part of the valve column and result in a decrease in the number of valve plates to the withstand voltage, which increases the withstand voltage of a single valve plate. In order to analyze the internal temperature distribution of the defect arrester under the action of overvoltage, a corresponding model with the electric tree defect was established.

The defective valve plates found in the disassembly inspection are all near the high-voltage side. For simulating the most serious defective situation, the dielectric constant of the defective valve plate was set to a larger value during the simulation such that the

defective valve plate had better conductivity, which was equivalent to the defective valve plate being short-circuited. The defective valve plate is placed on the high-voltage side, and the quantity is set to 1, 2 and 3.

### 4.5. Validation of the Model

The continuous operating voltage with an RMS of 34 kV is applied to the arrester model. The current obtained by the simulation calculation is shown in the red curve in Figure 14, and the actual measured current under the same voltage is shown in the blue curve in Figure 14.

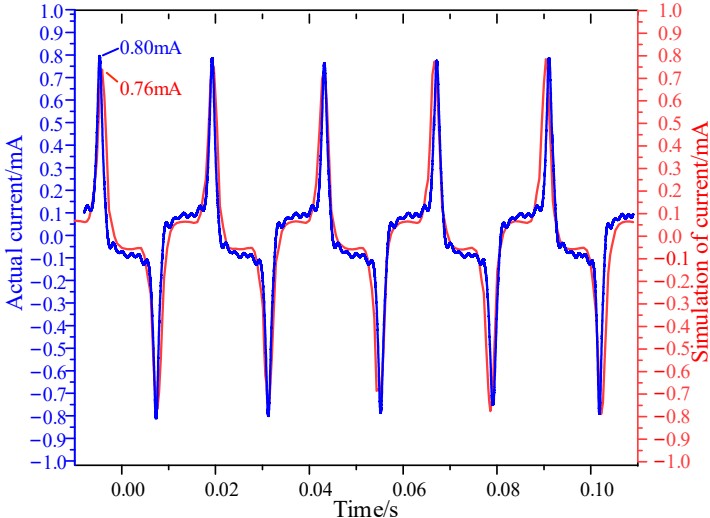

**Figure 14.** Simulation and actual current under 34 kV continuous voltage.

Figure 14 shows that the waveform of the simulated curve is close to that of the measured curve. The maximum peak value of the simulation current is 0.76 mA, and that of the actual current is 0.80 mA, which means that the error between them is 5% and that the parameter setting of the simulation model is reasonable.

To further verify the validity of the model, the temperature variation of channel 4 of the arrester was selected to compare with the simulation results, as shown in Figure 15.

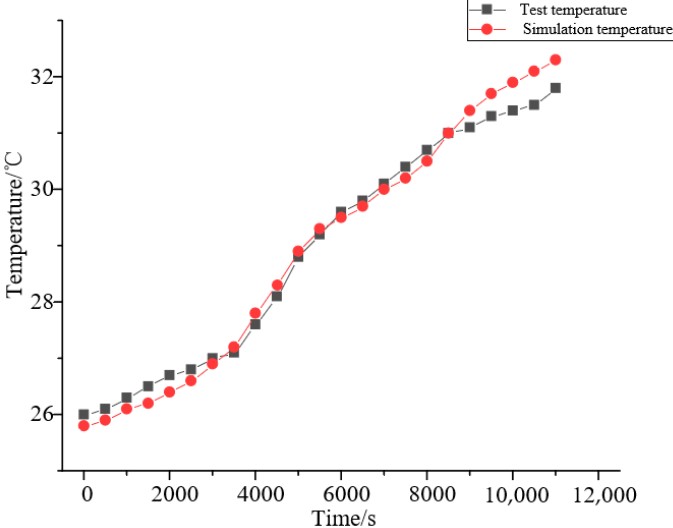

**Figure 15.** Temperature rise comparison between the measured and simulated results.

It can be seen from Figure 15 that the internal simulation and test temperature of the arrester gradually increased with time. The final actual temperature is slightly less than the

field-calculated temperature. The test and simulation maximum temperature are 31.7 °C and 32.3 °C, respectively.

### 4.6. *The Temperature Characteristics of the Arrester under Typical Overvoltage*

#### 4.6.1. Passing Section Overvoltage

The passing section overvoltage waveform shown in Figure 8a was imported into the simulation model, and the current and temperature distributions are shown in Figure 16.

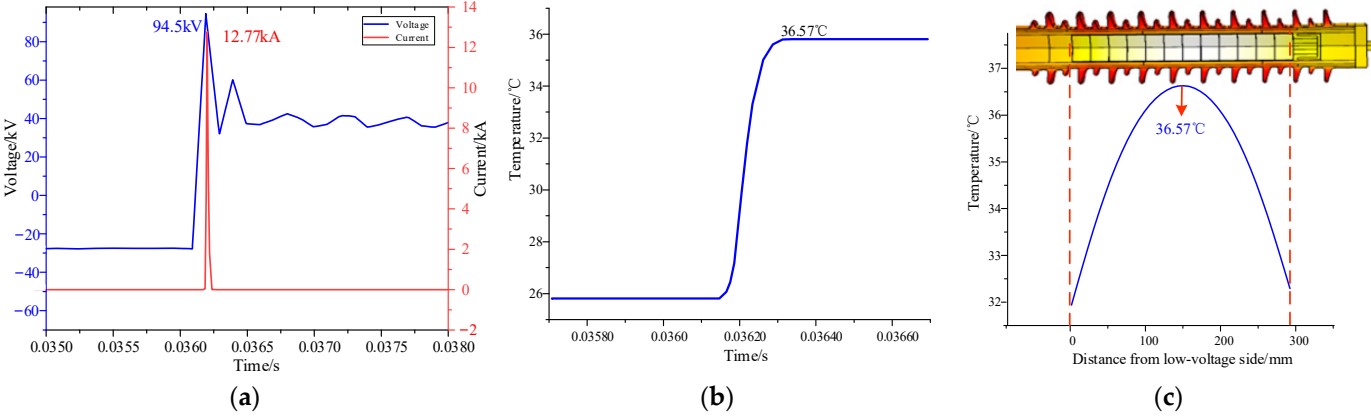

**Figure 16.** The current and temperature distributions of the arrester. (**a**) Passing section overvoltage and current. (**b**) Temperature variation of the arrester. (**c**) Temperature distribution of the valve column.

It can be seen from Figure 16 that, under the action of passing section overvoltage, the impulse current amplitude can reach 12.77 kA, which exceeds the nominal discharge current of 10 kA, and the temperature of the valve plate rises to 36.57 °C within 0.2 ms. This overvoltage is not enough to cause thermal collapse under normal conditions. The overall temperature of the valve column tends to be higher in the middle part, and the temperature on the low-voltage side is lower than that on the high-voltage side.

#### 4.6.2. High-Harmonic Overvoltage

The high-harmonic overvoltage waveform shown in Figure 8b was imported into the simulation model. Due to the duration being long, the overvoltage simulation time was set to 120 s. The simulation current and temperature distribution are shown in Figure 17.

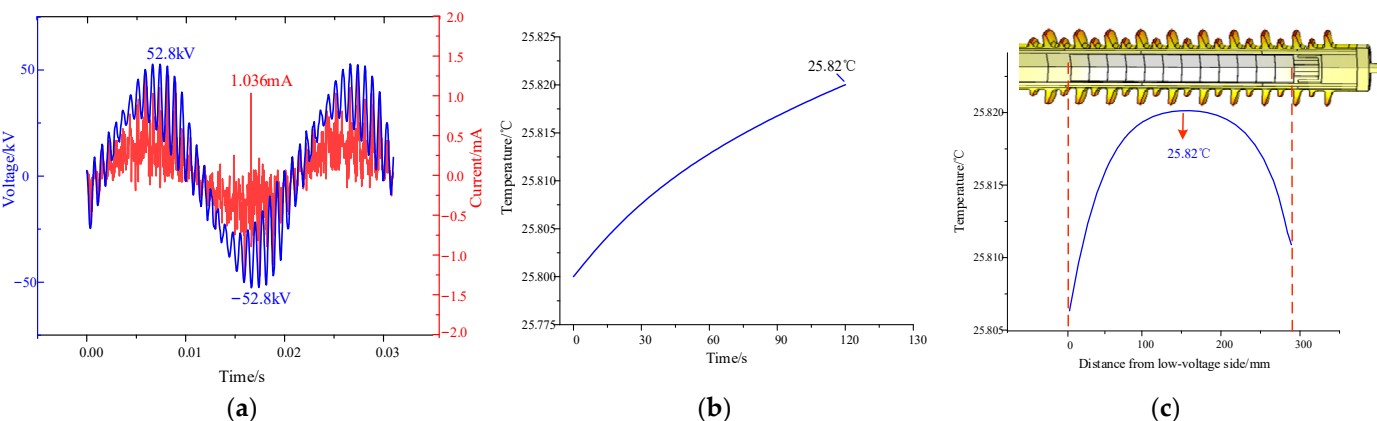

**Figure 17.** The current and temperature distribution of the arrester. (**a**) High-harmonic overvoltage and current. (**b**) Temperature variation of the arrester. (**c**) Temperature distribution of the valve column.

As shown in Figure 17, the temperature under the influence of high-harmonic overvoltage only rises by 0.02 °C within 120 s. The reason for this may be that the RMS of the high-harmonic voltage is relatively low, and the peak current flowing through the valve plate is only 1.036 mA, which belongs to the low-current working area of the arrester. However, the high-harmonic voltage has the characteristic of long duration. It may have a great impact on the temperature rise of the arrester when its effective value is high.

### 4.6.3. Steep Impulse Overvoltage

After the steep impulse overvoltage waveform shown in Figure 8c was imported into the simulation model, the results of the current and temperature distribution are shown in Figure 18.

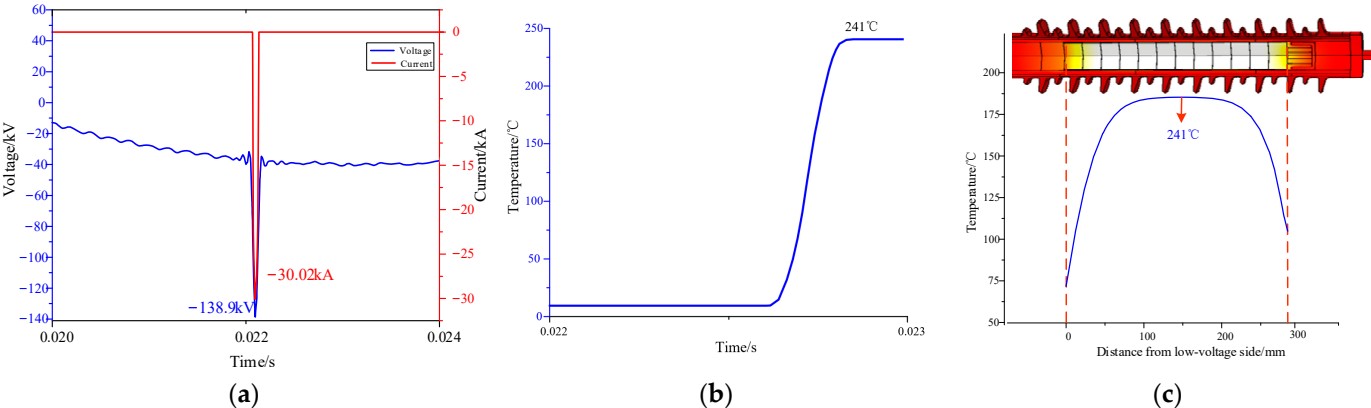

**Figure 18.** The current and temperature distribution of the arrester. (**a**) Steep impulse overvoltage and current. (**b**) Temperature variation of the arrester. (**c**) Temperature distribution of the valve column.

As shown in Figure 18, under the action of steep impulse overvoltage, the current amplitude of the arrester is 30.02 kA, which is much larger than the nominal discharge current of 10 kA, and the internal temperature of the arrester rises sharply in a short time—it can reach 241 °C.

### 4.7. Influence of the Defective Valve Plate Number on the Temperature Distribution of the Arrester

The influence of the defective valve plate number on the temperature rise under three typical overvoltage wave forms is also studied with the simulation, and the result is shown in Figure 19.

It can be seen from Figure 19 that, with the increase in the defective valve plate number, the internal temperature of the arrester shows an increasing trend under the three typical overvoltage wave forms, and the temperature is 44.75 °C, 25.86 °C and 536 °C, respectively, in the case of the most severe defects. The reason for this is that electric trees will reduce the insulation distance on the side of the valve plate, which will result in the rise of the withstand voltage of the rest valve plates. With the increase in the defective valve plate number, the abnormal temperature rise in the internal arrester may occur, even forming burst faults.

The influence of passing section overvoltage and steep impulse overvoltage is obvious, but the influence of high-harmonic overvoltage is small; this is because the effective value of high-harmonic overvoltage is small, only 28.9 kV, while the amplitudes of passing section overvoltage and steep impulse overvoltage are 94.5 kV and 138.9 kV, respectively. However, due to the long duration of the high-harmonic overvoltage, when its effective value is high, it may still cause a significant temperature rise of the arrester, resulting in thermal collapse [26–30].

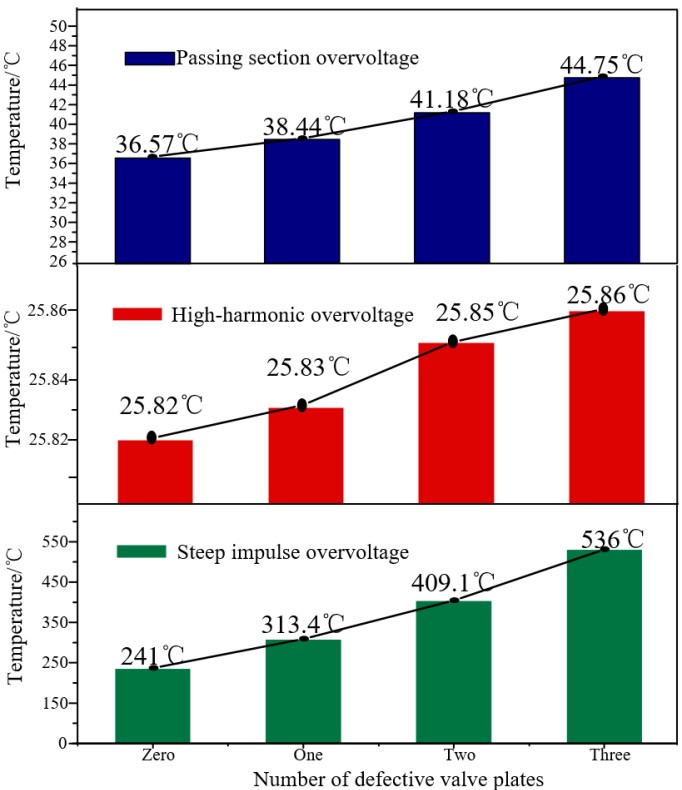

**Figure 19.** The influence of the defective valve plate number on the temperature rise.

*4.8. Influence of Air Velocity on the Temperature Distribution on the Arrester*

The high-speed airflow during the operation of EMU may have an influence on the temperature distribution of the arrester to some extent, which will affect the heat transfer between the arrester sheath and the outside air. The forced convective heat transfer of fluid around a cylinder follows the Zhukauskas formula [31].

$$
\begin{aligned}
N_u &= C R_e^n P_r^m \\
N_u &= \frac{hd}{\lambda} \\
R_e &= \frac{ud}{v}
\end{aligned}
\tag{7}
$$

where $N_u$ is the mean Nusselt number of the fluid around a cylinder, $R_e$ is the Reynolds number, $h$ is the convective heat transfer coefficient, $d$ is the diameter of the arrester, $\lambda$ is the thermal conductivity, $u$ is the air velocity, $v$ is the kinematic viscosity and $P_r$ is the Prandtl number. The values and simplified formulas of $C$, $n$ and $m$ are shown in Table 4.

**Table 4.** $C$, $n$ and $m$ and their simplified formula.

| Condition | $C$ | $n$ | $m$ | Simplified Formula |
|---|---|---|---|---|
| $5 < R_e < 10^3$ | 0.5 | 0.5 | 0.38 | $N_u = 0.44 R_e^{0.5}$ |
| $10^3 < R_e < 2 \times 10^5$ | 0.26 | 0.6 | 0.38 | $N_u = 0.22 R_e^{0.5}$ |
| $2 \times 10^5 < R_e < 2 \times 10^6$ | 0.023 | 0.8 | 0.37 | $N_u = 0.02 R_e^{0.5}$ |

When the ambient temperature is 20 °C, the thermal conductivity of the air is 0.0259 W/(m·K), the kinematic viscosity $v$ is $15.06 \times 10^{-6}$ m$^2$/s and the Prandtl number $P_r$ is 0.703. According to the simplified formula in Table 4 combined with the Zhukauskas formula, the convective heat transfer coefficient of airs with different velocities can be calculated, as shown in Table 5.

**Table 5.** Air heat transfer coefficient at different flow rates.

| The Air Velocity (km/h) | $R_e$ | $h$ (W/(m²·K)) |
|---|---|---|
| 50 | $1.1 \times 10^5$ | 50.3 |
| 150 | $3.3 \times 10^5$ | 112.2 |
| 250 | $5.5 \times 10^5$ | 168.8 |

It can be seen from the above analysis that the passing section overvoltage and steep impulse overvoltage have a high amplitude, and the probability of occurrence is also high; therefore, these two overvoltage forms are applied to the normal arrester. The internal temperature distribution of the arrester under three different velocities is calculated, and the results are shown in Figure 20.

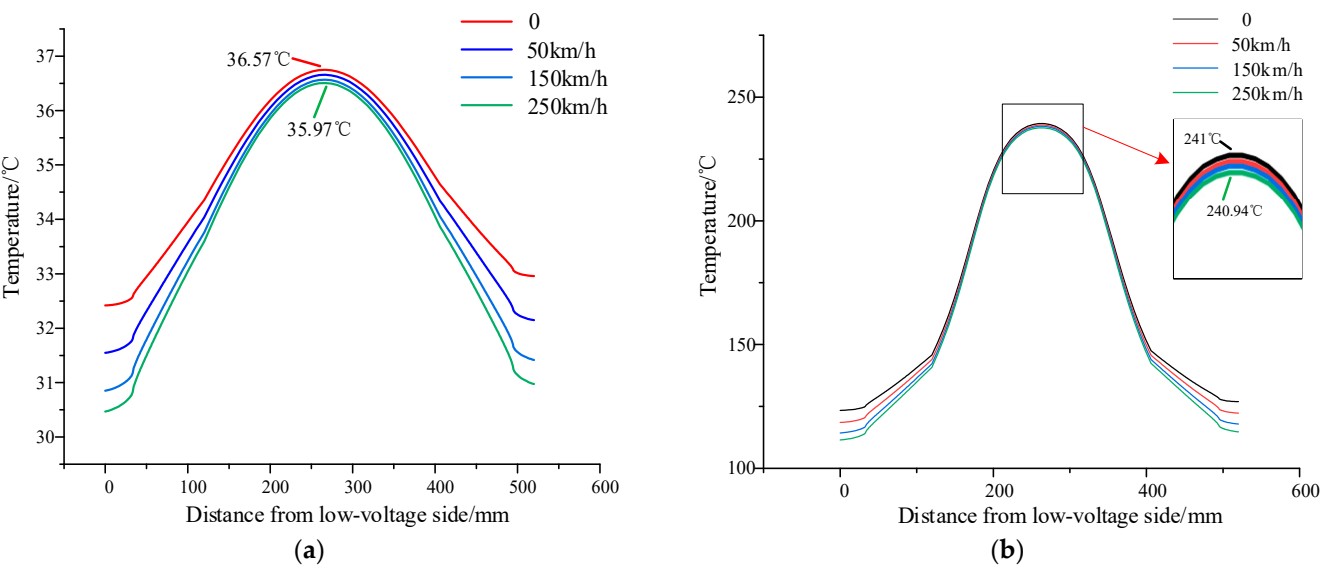

**Figure 20.** The temperature distribution of the arrester under different air velocities. (**a**) Passing section overvoltage. (**b**) Steep impulse overvoltage.

It can be seen from Figure 20 that, under the action of passing section overvoltage and steep impulse overvoltage, the maximum temperature of the valve plate is 36.57 °C and 241 °C, respectively, when the air is still, and it is 35.97 °C and 240.94 °C, respectively, when the air velocity is 250 km/h. Although the air velocity strengthens the convection heat dissipation, the temperature rise of the arrester is mainly caused by the heating of its internal valve column, which is wrapped in the sheath and other insulating materials, so the air velocity has little effect on its temperature. When the EMU is attacked by overvoltage during static or high-speed operation, the difference in temperature distribution is small.

## 5. Conclusions

Based on the experiments and simulation, this paper studied the influence of overvoltage on the internal temperature rise of the arrester and explained the relationship between the overvoltage and the arrester burst. The arrester internal temperature rise test was designed, and the internal temperature rise characteristics of the normal arrester under power frequency overvoltage and high-current impulse were obtained. An electrothermal coupling model of the arrester was established. Research on the internal temperature rise characteristics of the arrester under the action of typical overvoltage under different contaminations was carried out. The conclusions are as follows:

(1) After working at a continuous operating voltage for 3 h, the maximum temperature of the arrester valve plate increases by 5.2 °C. In the impulse current experiment, the maximum temperature of the arrester increases by 97.6 °C. The valve plates near the

two sides of the arrester dissipate heat faster and have a lower temperature. Applying multiple large currents continuously may cause harm to the roof arrester.

(2) The simulation results showed that, under the action of passing section overvoltage and steep impulse overvoltage, the internal temperature of the normal arrester reached 36.57 °C and 241 °C, respectively, in a short time. Under the action of high-harmonic overvoltage, the temperature rise was only 0.2 °C. This indicated that the high-amplitude overvoltage may be the main reason for the heat of the arrester.

(3) When the valve plate had electric tree defects, the internal temperature rise of the arrester was small under the action of high-harmonic overvoltage. Under the action of passing section overvoltage and steep impulse overvoltage, the temperature increases obviously with the number of defective valve plates, and the maximum temperature reaches 536 °C when the number of defective valve plates is 3.

(4) With the increase in air velocity, the internal temperature of the arrester decreased slightly under the passing section overvoltage and steep impulse overvoltage, so the EMU running at high speed has little influence on the internal temperature rise of the arrester under the action of overvoltage.

(5) The internal simulation temperature of the normal arrester and the defective arrester under the action of the steep impulse overvoltage was 8 times and 18 times that of the internal temperature of the arrester under the action of the power frequency overvoltage, respectively, which indicated that the roof overvoltage will not only accelerate the performance deterioration of the arrester but even lead to thermal collapse. The passing section overvoltage and steep impulse overvoltage have high amplitudes and many occurrences, so the roof overvoltage of the EMU is an important reason for the arrester burst.

**Author Contributions:** Conceptualization, S.W. and Q.O.; methodology, S.L. and H.L.; software, S.M.; validation, Q.O.; formal analysis, S.L.; investigation, Q.Z.; resources, J.L.; data curation, S.W.; writing—original draft preparation, Q.O. and S.W.; writing—review and editing, Q.O. and S.W.; visualization, Q.Z.; supervision, S.W.; project administration, F.L.; funding acquisition, J.L. All authors have read and agreed to the published version of the manuscript.

**Funding:** This research was funded by the Key R&D project of Hebei Province, grant number 19212109D, the National Key Research and Development Program of China, grant number 2018YFF01011903.

**Institutional Review Board Statement:** Not applicable.

**Informed Consent Statement:** Not applicable.

**Data Availability Statement:** The data are contained within the article. The data presented in this study are available on request from the corresponding author.

**Conflicts of Interest:** The authors declare no conflict of interest.

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
