# Peer review of "Study on the Measurement of the On-Site Overvoltage and Internal Temperature Rise Simulation of the EMU Arrester"

_applsci, doi:10.3390/app12157925_

Round 1

Reviewer 1 Report

The introduction is to be extended by adding some recently published papers related to the subject.

More details on the statistical analysis are to be added.

Are all the temperature measurements performed at the same surrounding temperatures?

How were the positions of the thermocouples chosen?

The governing equations used for the simulation are to be presented.

What is the used software?

Have you considered a 3D configuration?

For details on the numerical method are to be added.

A figure presenting the used mesh is to be added.

Why the heat transfer coefficient is fixed at 10 10 W/(m2K)

The assumptions used for the simulation are to be presented and justified.

The title of table 5 is to be checked.

Are the considered air velocity values realistic ?

A comparison between the experimental and numerical results is to be performed.

Reviewer 2 Report

The authors presented a study on the measurement of on-site overvoltage and simulated internal temperature rise of EMU arrester. The authors introduced the research topic and surveyed relevant literature. The authors established technical background backing up their study, upon which they developed and validated a simulation model. The model validation result they presented suggested that the model performs as intended. The authors carried out a range of simulations considering different fault conditions and, in each case, vividly showed and explained their simulation results. I must also point out that the report was well arranged.

However, the authors failed to specify the study's aims and objectives precisely. Establishing those aims and objectives is essential because they will guide me, the reviewer (and any other reader) in judging the relevance and quality of every other part of the report, including the methodology adopted, the design of experiments, results, explanations, and conclusions. Therefore, I’ll encourage the authors to restructure the manuscript and specify precisely what is/are the problem(s) they are trying to solve and how relevant are they.

Also, the use of English grammar in the report was poor, especially in the Abstract and the introduction sections. I’ve highlighted a few [but not all] of the areas that need their attention in the attached file.  I’ll encourage the authors to employ the services of a professional English grammar editor to improve the report's language for ease of reading and comprehension.

Round 2

Reviewer 1 Report

Can be accepted for publication

Author Response

Thanks very much for taking your time to review this manuscript again. We really appreciate all your comments and suggestions!

We read through the whole manuscript carefully, modify the inappropriate expressions in the manuscript, and correct some wrong words and non-standard expressions in the manuscript.

Reviewer 2 Report

I commend the authors for speedily revising the manuscript and responding to the reviewers’ observations as much as they have done. Such a speed response is a great asset in today’s fast-paced technological advancements. Also, I like to state that the authors did a good job of establishing the technical

Meanwhile, I want the authors to know that the purpose of this review is to enhance the quality of the manuscript and give the readers value for their time. Accordingly, I’m sorry to state that my concerns for a clearly defined objective(s) for this manuscript have not been addressed. The natural progression of a scientific report like this is as follows:

Introduce the study and give the background of problem(s) you’re trying to solve. Give your literature survey, showing what other researchers have done so far around the problem, showing their lines of thought and the different approaches they took to solving the problem. Then, establish the limitations to what other researchers have done that you intend to enhance, or state a new approach you intend to propose towards solving the problem. Clearly state this enhancement or the newly proposed approach as the objective of the manuscript, and every other thing (technical background, methodology, simulations, results, observations, conclusions, etc) in the manuscript revolves around it.

Owing to the above description of a scientific report, I want the authors to:

·         Clearly specify the problem(s) the study aimed to solve

·         Structure the whole report to reflect the lines of thought and the tasks performed towards solving the problem(s), and the results achieved.

·         Write a conclusion that shows that the proposed solution(s) was/were successful or unsuccessful.
